# NEURAL TOPIC MODELING
# WITH EMBEDDING CLUSTERING REGULARIZATION

## ABSTRACT

Topic models have been prevalent for decades with various applications like automatic text analysis due to their effectiveness and interpretability. However, existing topic models commonly suffer from the notorious topic collapsing issue: the discovered topics semantically collapse towards each other, leading to highly repetitive topics, insufficient topic discovery, and damaged model interpretability. In this paper, we propose a new neural topic model, Embedding Clustering Regularization Topic Model (ECRTM), to solve the topic collapsing issue. In addition to the reconstruction error of existing work, we propose a novel Embedding Clustering Regularization (ECR), which forces each topic embedding to be the center of a separately aggregated word embedding cluster in the semantic space. Instead of collapsing together, this makes topic embeddings away from each other and cover different semantics of word embeddings. Thus our ECR enables each produced topic to contain distinct word semantics, which alleviates topic collapsing. Through jointly optimizing our ECR objective and the neural topic modeling objective, ECRTM generates diverse and coherent topics together with high-quality topic distributions of documents. Extensive experiments on benchmark datasets demonstrate that ECRTM effectively addresses the topic collapsing issue and consistently surpasses state-of-the-art baselines in terms of topic quality, topic distributions of documents, and downstream classification tasks.

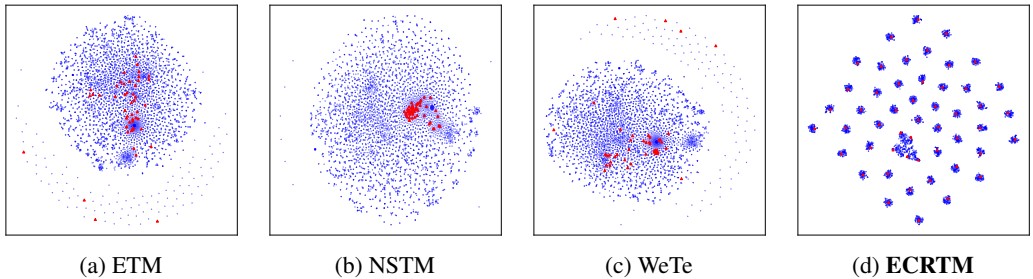

|  (a) ETM  |  (b) NSTM  |  (c) WeTe  |  (d) **ECRTM**  |
| --- | --- | --- | --- |

Figure 1: t-SNE visualization (van der Maaten & Hinton, 2008) of word embeddings (•) and topic embeddings (▲) under 50 topics. These show that while the topic embeddings mostly collapse together in previous state-of-the-art models (ETM (Dieng et al., 2020), NSTM (Zhao et al., 2021b), and WeTe (Wang et al., 2022)), our ECRTM successfully avoids the collapsing by forcing each topic embedding to be the center of a separately aggregated word embedding cluster.

## 1 INTRODUCTION

Topic models have achieved great success in document analysis via discovering latent semantics. They have various downstream applications (Boyd-Graber et al., 2017), like content recommendation (McAuley & Leskovec, 2013), summarization (Ma et al., 2012), and information retrieval (Wang et al., 2007). Current topic models can be roughly classified as two lines: conventional topic models based on probabilistic graphical models (Blei et al., 2003) or matrix factorization (Kim et al., 2015; Shi et al., 2018) and neural topic models (Miao et al., 2016; 2017; Srivastava & Sutton, 2017).

However, despite the current achievements, existing topic models commonly suffer from *topic collapsing*: the discovered topics tend to semantically collapse towards each other (Srivastava & Sutton, 2017), as exemplified in Table 1. We see these collapsed topics include many uninformative and repetitive words. Topic collapsing brings about several issues: (**1**) Topic collapsing results in

| | |
|---|---|
| Topic#1: | just show even come time one good really going know |
| Topic#2: | just even really something come going like actually things get |
| Topic#3: | just one even something come way really like always good |
| Topic#4: | just get going come one know even really something way |
| Topic#5: | just like inside get even look come one everything away |

Table 1: Top related words of the discovered topics by NSTM (Zhao et al., 2021b) on IMDB. These topics semantically collapse towards each other with many uninformative and repetitive words. Repetitions are underlined.

highly repetitive topics, which are less useful for downstream applications (Nan et al., 2019; Wu et al., 2020b). (**2**) Topic collapsing incurs insufficient topic discovery. Many latent topics are undisclosed, making the topic discovery insufficient to understand documents (Dieng et al., 2020). (**3**) Topic collapsing damages the interpretability of topic models. It becomes difficult to infer the real underlying topics that a document contains (Huynh et al., 2020). In consequence, topic collapsing impedes the utilization and extension of topic models; therefore it is crucial to overcome this challenge for building reliable topic modeling applications.

To address the topic collapsing issue and achieve robust topic modeling, we in this paper propose a novel neural topic model, **Embedding Clustering Regularization Topic Model** (**ECRTM**). First, we illustrate the reason for topic collapsing. Figure 1 shows topic embeddings mostly collapse together in the semantic space of previous state-of-the-art methods. This makes discovered topics contain similar word semantics and thus results in the topic collapsing. Then to avoid the collapsing of topic embeddings, we propose the novel Embedding Clustering Regularization (ECR) in addition to the reconstruction error of previous work. ECR considers topic embedding as cluster centers and word embeddings as cluster samples. To effectively regularize them, ECR models the clustering soft-assignments between them by solving a specifically defined optimal transport problem on them. As such, ECR forces each topic embedding to be the center of a separately aggregated word embedding cluster. Instead of collapsing together, this makes topic embeddings away from each other and cover different semantics of word embeddings. Thus our ECR enables each produced topic to contain distinct word semantics, which alleviates topic collapsing. Regularized by ECR, our ECRTM achieves robust topic modeling performance by producing diverse and coherent topics together with high-quality topic distributions of documents. Figure 1d shows the effectiveness of ECRTM. We conclude the main contributions of our paper as follows:

- We propose a novel embedding clustering regularization that avoids the collapsing of topic embeddings by forcing each topic embedding to be the center of a separately aggregated word embedding cluster, which effectively mitigates topic collapsing.

- We further propose a new neural topic model that jointly optimizes the topic modeling objective and the embedding clustering regularization objective. Our model can produce diverse and coherent topics with high-quality topic distributions of documents at the same time.

- We conduct extensive experiments on benchmark datasets and demonstrate that our model effectively addresses the topic collapsing issue and surpasses state-of-the-art baseline methods with substantially improved topic modeling performance.

## 2 RELATED WORK

**Conventional Topic Models** Conventional topic models (Hofmann, 1999; Blei et al., 2003; Blei & Lafferty, 2006; Mimno et al., 2009; Boyd-Graber et al., 2017) commonly employ probabilistic graphical models to model the generation process of documents with topics as latent variables. They infer model parameters with MCMC methods like Gibbs Sampling (Steyvers & Griffiths, 2007) or Variational Inference (Blei et al., 2017). Another line of work uses matrix factorization to model topics (Yan et al., 2013; Kim et al., 2015; Shi et al., 2018). Conventional topic models usually need model-specific derivations for different modeling assumptions.

**Neural Topic Models** Due to the success of Variational AutoEncoder (VAE, Kingma & Welling, 2014; Rezende et al., 2014), several neural topic models have been proposed (Miao et al., 2016;

2017; Srivastava & Sutton, 2017; Burkhardt & Kramer, 2019; Wu et al., 2020a; 2021; Dieng et al., 2019; 2020; Meng et al., 2020; Zhao et al., 2021a). Neural topic models commonly optimize the KL divergence and the reconstruction error between input and output by gradient back-propagation. Some studies directly cluster pre-trained word or sentence embeddings to produce topics (Sia et al., 2020; Zhang et al., 2022), but they are *not* topic models since they cannot infer the topic distributions of documents as required. Recent NSTM (Zhao et al., 2021b) and WeTe (Wang et al., 2022) measure the reconstruction error with optimal (conditional) transport distance. However, despite the different reconstruction error measurements, they still suffer from the topic collapsing, producing repetitive and less useful topics (see experiments in Sec. 4.2). Different from these, our proposed model aims to address the topic collapsing issue and substantially improve topic modeling performance. In addition to the reconstruction error of these previous work, our model proposes a novel embedding clustering regularization that avoids the collapsing of topic embeddings by forcing each topic embedding to be the center of a separately aggregated word embedding cluster. Then our model learns topics under this effective regularization and particularly addresses the topic collapsing issue.

## 3 METHODOLOGY

### 3.1 PROBLEM SETTING AND NOTATIONS

We recall the problem setting of topic modeling following LDA (Blei et al., 2003). Consider a document collection $\mathbf{X}$ with $V$ unique words (vocabulary size), and each document is denoted as $\mathbf{x}$. We require to discover $K$ topics from this document collection. The $k$-th latent topic is defined as a distribution over all words (topic-word distribution), denoted as $\boldsymbol{\beta}_k \in \mathbb{R}^V$. We have $\boldsymbol{\beta} = (\boldsymbol{\beta}_1, \ldots, \boldsymbol{\beta}_K) \in \mathbb{R}^{V \times K}$ as the topic-word distribution matrix of all topics. The topic distribution of a document (doc-topic distribution) refers to what topics it contains, noted as $\boldsymbol{\theta} \in \Delta_K$. [1]

### 3.2 WHAT CAUSES TOPIC COLLAPSING?

Despite the current achievements, most topic models suffer from *topic collapsing*: topics semantically collapse towards each other (see Table 1). We illustrate what causes topic collapsing by analyzing a kind of recently proposed state-of-the-art neural topic models (Dieng et al., 2020; Zhao et al., 2021b). These models compute the topic-word distribution matrix with two parameters: $\boldsymbol{\beta} = \mathbf{W}^\top \mathbf{T}$. Here $\mathbf{W} = (\mathbf{w}_1, \ldots, \mathbf{w}_V) \in \mathbb{R}^{D \times V}$ are the embeddings of $V$ words and $\mathbf{T} = (\mathbf{t}_1, \ldots, \mathbf{t}_K) \in \mathbb{R}^{D \times K}$ are the embeddings of $K$ topics, all in the same $D$-dimensional semantic space. They can facilitate learning by initializing $\mathbf{W}$ with pre-trained embeddings like GloVe (Pennington et al., 2014).

However, topic collapsing commonly happens in these models. We believe the reason lies in that their reconstruction error minimization incurs the collapsing of topic embeddings. Specifically, these models learn topic and word embeddings by minimizing the reconstruction error between topic distribution $\boldsymbol{\theta}$ and word distribution $\mathbf{x}$ of a document. For example, to measure reconstruction error, ETM (Dieng et al., 2020) uses expected log-likelihood, and recent NSTM (Zhao et al., 2021b) and WeTe (Wang et al., 2022) use optimal (conditional) transport distance. In fact, words in a document collection commonly are long-tail distributed following Zipf's law (Reed, 2001; Piantadosi, 2014)—roughly speaking, few words are high frequency and most are low frequency. Therefore the reconstruction is biased as it mainly reconstructs high-frequency words regardless of the reconstruction error measurements. Since topic and word embeddings are learned by minimizing reconstruction error, this biased reconstruction pushes most topic embeddings close to the embeddings of these high-frequency words in the semantic space. As a result, topic embeddings collapse together as shown in Figure 1. The topic-word distributions become similar to each other, making discovered topics contain similar word semantics. This leads to topic collapsing.

### 3.3 WHAT IS AN EFFECTIVE CLUSTERING REGULARIZATION ON EMBEDDINGS?

In this section, we explore how to design an effective clustering regularization to address the topic collapsing issue. Our analysis in Sec. 3.2 indicates topic collapsing happens because the reconstruction error minimization incurs the collapsing of topic embeddings in existing work. To solve

---

[1]Here $\Delta_K$ denotes a probability simplex $\Delta_K = \left\{ \boldsymbol{\theta} \in \mathbb{R}_+^K \,|\, \sum_{k=1}^K \theta_k = 1 \right\}$.

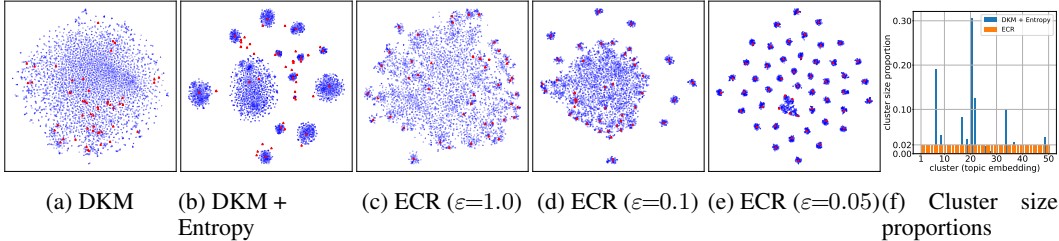

|                |                          |                 |                 |                   |                   |
|----------------|--------------------------|-----------------|-----------------|-------------------|-------------------|
| (a) DKM        | (b) DKM + Entropy        | (c) ECR ($\varepsilon$=1.0) | (d) ECR ($\varepsilon$=0.1) | (e) ECR ($\varepsilon$=0.05) | (f) Cluster size proportions |

Figure 2: t-SNE visualization (a-e) of word embeddings (•) and topic embeddings (▲) under 50 topics ($K$=50). (a): DKM cannot form separately aggregated clusters. (b): DKM + Entropy forms clusters but has a trivial solution that clusters of most topic embeddings are empty. (c,d,e): Our ECR forms clusters and also avoids the trivial solution of empty clusters with a small $\varepsilon$. (f): This quantitatively shows that while most cluster size proportions are zero in DKM + Entropy, our ECR successfully avoids this trivial solution with fulfilled preset cluster size constraints. Here all cluster size proportions of topic embeddings are preset equal as $1/K$=0.02.

this issue, we propose to design a clustering regularization in addition to the reconstruction error of existing work. Considering topic embeddings as cluster centers and word embeddings as cluster samples, we require the regularization to force each topic embedding to be the center of a separately aggregated word embedding cluster. So instead of collapsing together, topic embeddings will be away from each other and cover different semantics of word embeddings in the space. This will make each discovered topic contain distinct word semantics and thus alleviate topic collapsing. However, it is *non-trivial* to achieve such an effective clustering regularization, and we explore its requirements as follows.

**Supporting Joint Optimization**    As we regularize on neural topic models, we require the clustering regularization to support joint optimization on topic and word embeddings along with a neural topic modeling objective. Some studies (Sia et al., 2020) apply classical clustering algorithms, *e.g.*, KMeans and GMM, to produce topics by clustering pre-trained word embeddings. We clarify that they are *not* topic models as they only produce topics and cannot learn doc-topic distributions as required (but we compare them in experiments). We do not adopt these classical clustering algorithms and some other work (Song et al., 2013; Huang et al., 2014; Xie et al., 2016; Hsu & Lin, 2017; Yang et al., 2017) as our clustering regularization, because we cannot jointly optimize them along with a neural topic modeling objective.

**Producing Sparse Soft-assignments**    We also require the clustering regularization to produce sparse soft-assignments. Even supporting joint optimization, existing clustering methods may still lead to topic collapsing. For example, we propose to employ the state-of-the-art deep clustering method, Deep KMeans (DKM, Fard et al., 2020) that supports joint optimization. The clustering objective of DKM is to minimize the total Euclidean distance between centers and samples weighted by soft-assignments. We use DKM as a clustering regularization on topic and word embeddings:

$$\min_{\mathbf{W},\mathbf{T},\mathbf{p}} \sum_{j=1}^{V} \sum_{k=1}^{K} \|\mathbf{w}_j - \mathbf{t}_k\|^2 p_{jk}, \quad \text{where} \quad p_{jk} = \frac{e^{-\|\mathbf{w}_j - \mathbf{t}_k\|^2/\tau}}{\sum_{k'=1}^{K} e^{-\|\mathbf{w}_j - \mathbf{t}_{k'}\|^2/\tau}}. \tag{1}$$

Here $p_{jk}$ denotes the clustering soft-assignment of word embedding $\mathbf{w}_j$ assigned to topic embedding $\mathbf{t}_k$, which is modeled as a softmax function of the Euclidean distance between $\mathbf{w}_j$ and all topic embeddings ($\tau$ is a temperature hyper-parameter). Unfortunately, DKM still incurs topic collapsing (quantitative results are in Sec. 4.3). We see from Figure 2a that DKM cannot form separately aggregated clusters, so the topic embeddings (centers) cannot be separated but collapse together. To solve this issue, we require the clustering regularization to produce sparse soft-assignments—each word embedding is mainly assigned to only one topic embedding and rarely to others, which pushes each word embedding only close to one topic embedding and away from all others in the semantic space. This way can form separately aggregated word embedding clusters with topic embeddings as centers, which encourages topic embeddings to be away from each other.

**Fulfilling Preset Cluster Size Constraints**    We further require the clustering regularization to fulfill preset cluster size constraints. Only producing sparse soft-assignments may still result in topic

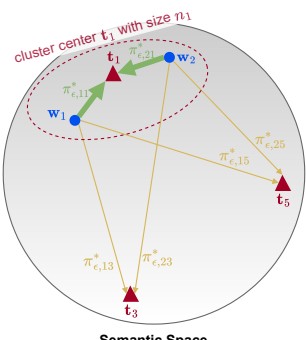

**Algorithm 1** Training algorithm for ECRTM.

**Input**: document collection $\mathbf{X}$, preset cluster size constraint $\mathbf{s}$, number of epochs $n_{\text{epoch}}$;

**Output**: model parameters $\Theta, \mathbf{W}, \mathbf{T}$;

1: **for** 1 to $n_{\text{epoch}}$ **do**
2:   **for** each mini-batch $(\mathbf{x}^{(1)}, \mathbf{x}^{(2)}, \dots, \mathbf{x}^{(N)})$ from $\mathbf{X}$ **do**
3:     // Sinkhorn's algorithm;
4:     $C_{jk} = \|\mathbf{w}_j - \mathbf{t}_k\|^2 \quad \forall\, j, k$;
5:     $\mathbf{M} = \exp(-\mathbf{C}/\epsilon)$;
6:     Initialize $\mathbf{b} \leftarrow \mathbb{1}_K$;
7:     **while** not converged and not reach max iterations **do**
8:       $\mathbf{a} \leftarrow \frac{1}{V}\frac{\mathbb{1}_V}{\mathbf{M}\mathbf{b}}$ , $\mathbf{b} \leftarrow \frac{\mathbf{s}}{\mathbf{M}^\top \mathbf{a}}$;
9:     **end while**
10:    Compute $\boldsymbol{\pi}_\varepsilon^* \leftarrow \text{diag}(\mathbf{a})\mathbf{M}\text{diag}(\mathbf{b})$;
11:    Compute $\mathcal{L}_{\text{TM}} + \lambda_{\text{ECR}}\mathcal{L}_{\text{ECR}}$ (Eq. (6));
12:    Update $\Theta, \mathbf{W}, \mathbf{T}$ with a gradient step;
13:   **end for**
14: **end for**

Figure 3: Illustration of ECR. ECR clusters word embeddings $\mathbf{w}_j$ (●) as samples and topic embeddings $\mathbf{t}_k$ (▲) as centers with soft-assignment $\pi_{\epsilon,jk}^*$. The cluster size of center $\mathbf{t}_k$ is constrained as $n_k$. Here ECR pushes $\mathbf{w}_1$ close to $\mathbf{t}_1$ and away from $\mathbf{t}_3$ and $\mathbf{t}_5$. It is similar for $\mathbf{w}_2$.

collapsing. To make the soft-assignments sparse, we propose DKM + Entropy that jointly minimizes the entropy of soft-assignments, $\sum_{j=1}^{V}\sum_{k=1}^{K} -p_{jk}\log p_{jk}$ with the clustering objective of DKM (Eq. (1)). However, this way still leads to topic collapsing (quantitative results are in Sec. 4.3). Figure 2b shows this way indeed forms separately aggregated clusters for some topic embeddings, but unfortunately the clustering solution is trivial—the clusters of most topic embeddings are empty, as quantitatively shown in Figure 2f. As a result, the topic embeddings of these empty clusters cannot be separated to cover distinct semantics but only collapse to others in the space. To avoid such trivial solutions of empty clusters, we propose to preset constraints on the size of each cluster (must not be empty) and require the clustering regularization to fulfill these constraints.

### 3.4 EMBEDDING CLUSTERING REGULARIZATION

To meet the above requirements, we in this section introduce a novel method, **Embedding Clustering Regularization** (**ECR**). Figure 3 illustrates ECR, and Figure 2e shows its effectiveness.

**Presetting Cluster Size Constraints** We first preset the cluster size constraints to be fulfilled to avoid trivial solutions of empty clusters. Recall that we have $K$ topic embeddings as centers and $V$ word embeddings as samples. We denote the cluster size of topic embedding $\mathbf{t}_k$ as $n_k$ and the cluster size proportion as $s_k = n_k/V$. We have $\mathbf{s} = (s_1, \dots, s_K)^\top \in \Delta_K$ as the vector of all cluster size proportions. In our experiments, we set all cluster sizes as equal, $n_k = V/K$, so $\mathbf{s}$ is uniform, $\mathbf{s} = (1/K, \dots, 1/K)^\top$. This is because we often lack prior knowledge about cluster sizes, and it is reasonable to assume each cluster (topic) includes the same amount of semantic information without loss of generality. [2] Experiments also show this assumption works well (see Sec. 4.2).

**Embedding Clustering Regularization (ECR)** To meet the above requirements, we propose ECR that models clustering soft-assignments with the transport plan of a specifically defined optimal transport problem. Specifically, we define two discrete measures of topic ($\mathbf{t}_k$) and word embeddings ($\mathbf{w}_j$): $\gamma = \sum_{j=1}^{V} \frac{1}{V}\delta_{\mathbf{w}_j}$ and $\phi = \sum_{k=1}^{K} s_k \delta_{\mathbf{t}_k}$, where $\delta_x$ denotes the Dirac unit mass on $x$. We formulate the entropic regularized optimal transport problem between $\gamma$ and $\phi$ as

$$\arg\min_{\boldsymbol{\pi} \in \mathbb{R}_+^{V \times K}} \mathcal{L}_{\text{OT}_\varepsilon}(\gamma, \phi), \quad \mathcal{L}_{\text{OT}_\varepsilon}(\gamma, \phi) = \sum_{j=1}^{V}\sum_{k=1}^{K} \|\mathbf{w}_j - \mathbf{t}_k\|^2 \pi_{jk} + \sum_{j=1}^{V}\sum_{k=1}^{K} \varepsilon\pi_{jk}(\log(\pi_{jk}) - 1),$$

$$\text{s.t.} \quad \boldsymbol{\pi}\mathbb{1}_K = \frac{1}{V}\mathbb{1}_V \quad \text{and} \quad \boldsymbol{\pi}^\top\mathbb{1}_V = \mathbf{s}. \tag{2}$$

---

[2] Note that $\mathbf{s}$ can be determined by prior knowledge from experts, and we leave this as future work.

Here the first term is the original optimal transport problem, and the second term is the entropic regularization with hyper-parameter $\varepsilon$ to make this problem tractable (Canas & Rosasco, 2012). Eq. (2) is to find the optimal transport plan $\boldsymbol{\pi}_\varepsilon^*$ that minimizes the total cost of transporting weight from word embeddings to topic embeddings. We measure the transport cost between word embedding $\mathbf{w}_j$ and topic embedding $\mathbf{t}_k$ by Euclidean distance: $C_{jk} = \|\mathbf{w}_j - \mathbf{t}_k\|^2$, and the transport cost matrix is $\mathbf{C} \in \mathbb{R}^{V \times K}$. The two conditions in Eq. (2) restrict the weight of each word embedding $\mathbf{w}_j$ as $\frac{1}{V}$ and each topic embedding $\mathbf{t}_k$ as $s_k$, where $\mathbb{1}_K$ ($\mathbb{1}_V$) is a $K$ ($V$) dimensional column vector of ones. $\pi_{jk}$ denotes the transport weight from $\mathbf{w}_j$ to $\mathbf{t}_k$; $\boldsymbol{\pi} \in \mathbb{R}_+^{V \times K}$ is the transport plan that includes the transport weight of each word embedding to fulfill the weight of each topic embedding.

To meet the requirements of an effective clustering regularization, we model clustering soft-assignments with the optimal transport plan $\boldsymbol{\pi}_\varepsilon^*$: the soft-assignment of $\mathbf{w}_j$ to $\mathbf{t}_k$ is the transport weight between them, $\pi_{\varepsilon,jk}^*$. Then we formulate our ECR objective by minimizing the total distance between word and topic embeddings weighted by these soft-assignments:

$$\mathcal{L}_{\text{ECR}} = \sum_{j=1}^{V} \sum_{k=1}^{K} \|\mathbf{w}_j - \mathbf{t}_k\|^2 \pi_{\varepsilon,jk}^*, \text{ where } \boldsymbol{\pi}_\varepsilon^* = \text{sinkhorn}(\gamma, \phi, \varepsilon) \approx \underset{\boldsymbol{\pi} \in \mathbb{R}_+^{V \times K}}{\arg\min} \mathcal{L}_{\text{OT}_\varepsilon}(\gamma, \phi). \quad (3)$$

Here we compute $\boldsymbol{\pi}_\varepsilon^*$ through Sinkhorn's algorithm (Sinkhorn, 1964; Cuturi, 2013, Algorithm 1), a fast iterative scheme suited to the execution of GPU (Peyré et al., 2019). By doing so, $\boldsymbol{\pi}_\varepsilon^*$ is a differentiable variable parameterized by transport cost matrix $\mathbf{C}$ (Salimans et al., 2018; Genevay et al., 2018). Due to this, minimizing $\pi_{\varepsilon,jk}^*$ increases transport cost $C_{jk}$, *i.e.*, the distance between $\mathbf{w}_j$ and $\mathbf{t}_k$; otherwise maximizing it decreases the distance (Genevay et al., 2019). Thus we can exactly model $\boldsymbol{\pi}_\varepsilon^*$ as differentiable clustering soft-assignments between topic and word embeddings.

**ECR is an Effective Clustering Regularization** As shown in Figures 2e and 2f, ECR is an effective clustering regularization that meets the requirements in Sec. 3.2. First, ECR supports joint optimization since $\boldsymbol{\pi}_\varepsilon^*$ is differentiable as aforementioned. Second, ECR produces sparse soft-assignments. It is proven that $\boldsymbol{\pi}_\varepsilon^*$ converges to the optimal solution of the original optimal transport problem when $\varepsilon \to 0$, which leads to a sparse transport plan (Peyré et al., 2019). Hence ECR (Eq. (3)) produces sparse soft-assignments under a small $\varepsilon$. With sparse soft-assignments, ECR pushes each word embedding only close to one topic embedding and away from all others, which forms separately aggregated clusters. We illustrate this property in Figures 2c to 2e. Last, ECR fulfills preset cluster size constraints. In Eq. (2), the transport plan is restricted by two conditions indicating the weight of each word embedding $\mathbf{w}_j$ is $\frac{1}{V}$ and each topic embedding $\mathbf{t}_k$ is $s_k$. These ensure the sparse optimal transport plan $\boldsymbol{\pi}_\varepsilon^*$ needs to transport $n_k$ word embeddings to topic embedding $\mathbf{t}_k$ to balance the weight, such that $n_k \times \frac{1}{V} = s_k$. Accordingly, ECR fulfills the preset cluster size constraints with $\boldsymbol{\pi}_\varepsilon^*$ as clustering soft-assignments, which avoids trivial clustering solutions of empty clusters as shown in Figure 2f. To sum up, our effective ECR forces each topic embedding to be the center of a separately aggregated word embedding cluster. This makes topic embeddings away from each other and cover different semantics of word embeddings in the space, which alleviates topic collapsing.

### 3.5 NEURAL TOPIC MODELING WITH EMBEDDING CLUSTERING REGULARIZATION

In this section, we propose a novel neural topic model, **Embedding Clustering Regularization Topic Model** (**ECRTM**) that jointly optimizes the topic modeling objective and our proposed ECR objective. Algorithm 1 shows the training algorithm of ECRTM.

**Inferring Topic Distributions** We devise the prior and variational distribution following VAE (Kingma & Welling, 2014) to infer topic distributions. Since it is difficult to directly apply the reparameterization trick of VAE on a Dirichlet distribution as in LDA, we employ a logistic normal prior to avoid this issue (Srivastava & Sutton, 2017). In detail, we draw a latent variable, $\mathbf{r}$, from a logistic normal distribution: $p(\mathbf{r}) = \mathcal{LN}(\boldsymbol{\mu}_0, \boldsymbol{\Sigma}_0)$, where $\boldsymbol{\mu}_0$ and $\boldsymbol{\Sigma}_0$ are the mean and diagonal covariance matrix. Then we use an encoder network that outputs parameters of the variational distribution, the mean vector $\boldsymbol{\mu} = f_{\boldsymbol{\mu}}(\mathbf{x})$ and covariance matrix $\boldsymbol{\Sigma} = \text{diag}(f_{\boldsymbol{\Sigma}}(\mathbf{x}))$. So the variational distribution is $q_{\Theta}(\mathbf{r}|\mathbf{x}) = \mathcal{N}(\boldsymbol{\mu}, \boldsymbol{\Sigma})$ where $\Theta$ denotes the parameters of $f_{\boldsymbol{\mu}}$ and $f_{\boldsymbol{\Sigma}}$. By applying the reparameterization trick (Kingma & Welling, 2014), we sample $\mathbf{r} = \boldsymbol{\mu} + \boldsymbol{\Sigma}^{1/2} \boldsymbol{\epsilon}$ where $\boldsymbol{\epsilon} \sim \mathcal{N}(0, \mathbf{I})$. We obtain the topic distribution $\boldsymbol{\theta}$ with a softmax function as $\boldsymbol{\theta} = \text{softmax}(\mathbf{r})$.

**Reconstructing Documents** We then reconstruct the input documents with topic-word distribution matrix $\boldsymbol{\beta} \in \mathbb{R}^{V \times K}$. Recall that $\boldsymbol{\beta}$ indicates the weights between all topics and words. Previous methods commonly model $\boldsymbol{\beta}$ as the product of topic and word embeddings. Differently, our model uses the proposed ECR as a clustering regularization on topic and word embeddings, so $\boldsymbol{\beta}$ also needs to reflect the learned clustering assignments between them. We do not directly model $\boldsymbol{\beta}$ with the soft-assignments $\boldsymbol{\pi}_\varepsilon^*$ of our ECR. This is because this way cannot provide sufficient weight information between topics and words for reconstruction since $\boldsymbol{\pi}_\varepsilon^*$ is very sparse as aforementioned (most values are close to zero). Experiments also show this way is ineffective. Hence we need a less sparse $\boldsymbol{\beta}$ for reconstruction. To this end, we propose to model $\boldsymbol{\beta}$ with the less sparse soft-assignments following DKM (Fard et al., 2020):

$$\beta_{jk} = \frac{e^{-\|\mathbf{w}_j - \mathbf{t}_k\|^2/\tau}}{\sum_{k'=1}^{K} e^{-\|\mathbf{w}_j - \mathbf{t}_{k'}\|^2/\tau}} \tag{4}$$

where $\tau$ is a temperature hyper-parameter. This formulation can reflect the learned clustering assignments between topic and word embeddings and is less sparse to provide sufficient information for reconstruction. With the topic distribution $\boldsymbol{\theta}$ and the topic-word distribution matrix $\boldsymbol{\beta}$, we routinely sample the reconstructed document from a Multinomial distribution $\mathrm{Multi}(\mathrm{softmax}(\boldsymbol{\beta}\boldsymbol{\theta}))$.

**Overall Objective Function** Given a mini-batch of $N$ documents $(\mathbf{x}^{(1)}, \ldots, \mathbf{x}^{(N)})$, we formulate the topic modeling objective function following VAE as

$$\mathcal{L}_{\mathrm{TM}} = \frac{1}{N} \sum_{i=1}^{N} -\mathbf{x}^{(i)\top} \log(\mathrm{softmax}(\boldsymbol{\beta}\boldsymbol{\theta}^{(i)})) + \mathbf{KL}\left[q_\Theta(\mathbf{r}^{(i)}|\mathbf{x}^{(i)})\|p(\mathbf{r}^{(i)})\right]. \tag{5}$$

The first term is the reconstruction error, and the second term is the KL divergence between the prior and variational distribution. ECRTM learns topics regularized by our ECR. We define the overall objective function of ECRTM as a combination of $\mathcal{L}_{\mathrm{TM}}$ (Eq. (5)) and $\mathcal{L}_{\mathrm{ECR}}$ (Eq. (3)):

$$\min_{\Theta, \mathbf{W}, \mathbf{T}} \mathcal{L}_{\mathrm{TM}} + \lambda_{\mathrm{ECR}} \mathcal{L}_{\mathrm{ECR}} \tag{6}$$

where $\lambda_{\mathrm{ECR}}$ is a weight hyper-parameter. This overall objective enables ECRTM to aggregate the embeddings of related words to form separate clusters with topic embeddings as centers and avoids the collapsing of topic embeddings. Thus our ECRTM can alleviate the topic collapsing issue and learn coherent and diverse topics together with high-quality doc-topic distributions at the same time.

## 4 EXPERIMENT

### 4.1 EXPERIMENT SETUP

**Datasets** We adopt the following benchmark document datasets for experiments: (i) 20 News Groups (**20NG**, Lang, 1995) is one of the most popular datasets for evaluating topic models, including news articles with 20 labels; (ii) **IMDB** (Maas et al., 2011) is the movie reviews containing two labels (positive and negative); (iii) **Yahoo Answer** (Zhang et al., 2015) is the question titles, contents, and the best answers from the Yahoo website with 10 labels. (iv) **AG News** (Zhang et al., 2015) contains news titles and descriptions, divided into 4 categories like Sports and Business. The preprocessing details are described in Appendix A.1.

**Evaluation Metrics** We evaluate topic models concerning **topic and doc-topic distribution quality** (perplexity is not considered since it disagrees with human interpretability (Chang et al., 2009; Hoyle et al., 2021)). For topic quality, we consider **Topic Coherence** and **Topic Diversity**. Topic coherence measures whether the top words of discovered topics are coherent (Newman et al., 2010; Wang & Blei, 2011). We employ the widely-used topic coherence metric, Coherence Value ($C_V$) (Röder et al., 2015). Besides, topic diversity measures the differences between discovered topics to verify if topic collapsing happens. We use Topic Diversity (TD, Dieng et al., 2020) to evaluate topic diversity, which computes the proportion of unique words in the discovered topics. Additionally, we conduct **document clustering tasks** to evaluate doc-topic distribution quality following Zhao et al. (2021b); Wang et al. (2022). We use the most significant topics in the topic distributions of testing documents as their document clustering assignments and adopt the commonly-used Purity and NMI (Manning et al., 2008) for evaluation. Note that our intention is not to achieve state-of-the-art document clustering results but compare doc-topic distribution quality.

Table 2: Topic quality of topic coherence ($C_V$) and topic diversity (TD) under 50 and 100 topics ($K=50$ and $K=100$). The best scores are in **bold**. ‡ means the gain of ECRTM is statistically significant at 0.05 level.

| Model | 20NG | | | | IMDB | | | |
| --- | --- | --- | --- | --- | --- | --- | --- | --- |
| | $K=50$ | | $K=100$ | | $K=50$ | | $K=100$ | |
| | $C_V$ | TD | $C_V$ | TD | $C_V$ | TD | $C_V$ | TD |
| LDA | ‡0.385±0.005 | ‡0.655±0.024 | ‡0.387±0.004 | ‡0.622±0.027 | ‡0.347±0.004 | ‡0.788±0.033 | ‡0.342±0.006 | ‡0.691±0.053 |
| KM | ‡0.251±0.008 | ‡0.204±0.017 | ‡0.294±0.013 | ‡0.317±0.017 | ‡0.213±0.008 | ‡0.219±0.011 | ‡0.244±0.003 | ‡0.302±0.006 |
| WLDA | ‡0.378±0.007 | ‡0.375±0.023 | ‡0.369±0.004 | ‡0.273±0.022 | ‡0.311±0.007 | ‡0.053±0.008 | ‡0.320±0.003 | ‡0.069±0.003 |
| ETM | ‡0.375±0.007 | ‡0.704±0.024 | ‡0.369±0.001 | ‡0.573±0.016 | ‡0.346±0.005 | ‡0.557±0.014 | ‡0.341±0.005 | ‡0.371±0.017 |
| NSTM | ‡0.395±0.005 | ‡0.427±0.017 | ‡0.391±0.003 | ‡0.473±0.025 | ‡0.334±0.003 | ‡0.175±0.010 | ‡0.340±0.003 | ‡0.255±0.012 |
| WeTe | ‡0.383±0.007 | ‡0.949±0.003 | ‡0.352±0.008 | ‡0.742±0.025 | ‡0.368±0.008 | ‡0.931±0.021 | ‡0.293±0.018 | ‡0.638±0.024 |
| **ECRTM** | **0.431±0.013** | **0.964±0.009** | **0.405±0.006** | **0.904±0.028** | **0.395±0.005** | **0.979±0.020** | **0.373±0.006** | **0.887±0.019** |

| Model | Yahoo Answer | | | | AG News | | | |
| --- | --- | --- | --- | --- | --- | --- | --- | --- |
| | $K=50$ | | $K=100$ | | $K=50$ | | $K=100$ | |
| | $C_V$ | TD | $C_V$ | TD | $C_V$ | TD | $C_V$ | TD |
| LDA | ‡0.359±0.007 | ‡0.843±0.012 | ‡0.359±0.005 | ‡0.602±0.030 | ‡0.364±0.006 | ‡0.864±0.027 | ‡0.349±0.005 | ‡0.696±0.018 |
| KM | ‡0.271±0.018 | ‡0.242±0.008 | ‡0.297±0.006 | ‡0.345±0.008 | ‡0.241±0.015 | ‡0.264±0.026 | ‡0.289±0.007 | ‡0.395±0.016 |
| WLDA | ‡0.333±0.007 | ‡0.322±0.024 | ‡0.338±0.006 | ‡0.292±0.021 | ‡0.384±0.004 | ‡0.410±0.018 | ‡0.378±0.005 | ‡0.323±0.035 |
| ETM | ‡0.354±0.003 | ‡0.719±0.022 | ‡0.353±0.002 | ‡0.624±0.024 | ‡0.364±0.002 | ‡0.819±0.017 | ‡0.371±0.005 | ‡0.773±0.005 |
| NSTM | ‡0.390±0.009 | ‡0.658±0.011 | 0.387±0.001 | ‡0.659±0.005 | ‡0.411±0.011 | ‡0.873±0.019 | **0.421±0.003** | ‡0.832±0.010 |
| WeTe | ‡0.367±0.010 | ‡0.878±0.020 | ‡0.353±0.007 | ‡0.544±0.019 | ‡0.383±0.007 | ‡0.945±0.004 | ‡0.363±0.005 | ‡0.827±0.023 |
| **ECRTM** | **0.405±0.004** | **0.985±0.013** | **0.389±0.006** | **0.903±0.033** | **0.466±0.012** | **0.961±0.009** | 0.416±0.003 | **0.981±0.012** |

Table 3: Document clustering of Purity and NMI under 50 and 100 topics ($K=50$ and $K=100$). The best scores are in **bold**. ‡ means the gain of ECRTM is statistically significant at 0.05 level.

| Model | 20NG | | | | IMDB | | | |
| --- | --- | --- | --- | --- | --- | --- | --- | --- |
| | $K=50$ | | $K=100$ | | $K=50$ | | $K=100$ | |
| | Purity | NMI | Purity | NMI | Purity | NMI | Purity | NMI |
| LDA | ‡0.367±0.024 | ‡0.364±0.016 | ‡0.364±0.011 | ‡0.346±0.005 | ‡0.614±0.006 | ‡0.041±0.008 | ‡0.600±0.009 | ‡0.037±0.007 |
| WLDA | ‡0.233±0.008 | ‡0.157±0.007 | ‡0.292±0.009 | ‡0.207±0.008 | ‡0.589±0.008 | ‡0.011±0.002 | ‡0.602±0.005 | ‡0.013±0.001 |
| ETM | ‡0.347±0.026 | ‡0.319±0.027 | ‡0.394±0.013 | ‡0.339±0.010 | ‡0.660±0.004 | ‡0.038±0.002 | ‡0.648±0.009 | ‡0.037±0.002 |
| NSTM | ‡0.354±0.032 | ‡0.356±0.015 | ‡0.383±0.021 | ‡0.363±0.007 | ‡0.658±0.008 | ‡0.040±0.002 | ‡0.659±0.010 | ‡0.039±0.003 |
| WeTe | ‡0.268±0.012 | ‡0.304±0.009 | ‡0.338±0.032 | ‡0.348±0.028 | ‡0.587±0.014 | ‡0.031±0.006 | ‡0.589±0.006 | ‡0.025±0.001 |
| **ECRTM** | **0.560±0.037** | **0.524±0.025** | **0.555±0.018** | **0.494±0.009** | **0.690±0.009** | **0.056±0.007** | **0.694±0.006** | **0.049±0.002** |

| Model | Yahoo Answer | | | | AG News | | | |
| --- | --- | --- | --- | --- | --- | --- | --- | --- |
| | $K=50$ | | $K=100$ | | $K=50$ | | $K=100$ | |
| | Purity | NMI | Purity | NMI | Purity | NMI | Purity | NMI |
| LDA | ‡0.288±0.017 | ‡0.144±0.013 | ‡0.297±0.019 | ‡0.148±0.007 | ‡0.640±0.028 | ‡0.193±0.017 | ‡0.654±0.012 | ‡0.194±0.006 |
| WLDA | ‡0.255±0.007 | ‡0.084±0.005 | ‡0.303±0.010 | ‡0.127±0.010 | ‡0.580±0.011 | ‡0.151±0.010 | ‡0.653±0.005 | ‡0.188±0.004 |
| ETM | ‡0.405±0.015 | ‡0.192±0.008 | ‡0.428±0.006 | ‡0.208±0.005 | ‡0.679±0.012 | ‡0.224±0.015 | ‡0.674±0.006 | ‡0.204±0.006 |
| NSTM | ‡0.395±0.022 | ‡0.241±0.009 | ‡0.405±0.013 | ‡0.242±0.011 | ‡0.719±0.041 | ‡0.324±0.022 | ‡0.764±0.018 | ‡0.359±0.008 |
| WeTe | ‡0.389±0.019 | ‡0.252±0.015 | ‡0.444±0.016 | ‡0.269±0.012 | ‡0.641±0.023 | ‡0.268±0.013 | ‡0.699±0.008 | ‡0.271±0.009 |
| **ECRTM** | **0.550±0.013** | **0.295±0.009** | **0.563±0.007** | **0.311±0.003** | **0.802±0.009** | **0.367±0.013** | **0.812±0.020** | **0.428±0.016** |

**Baseline Models** We consider the following state-of-the-art models for comparison: (i) **LDA** (Blei et al., 2003), one of the most widely-used probabilistic topic models; (ii) **KM** (Sia et al., 2020), directly clustering word embeddings to produce topics. Note that we cannot use it for document clustering since it does not infer the doc-topic distributions. (iii) **WLDA** (Nan et al., 2019), a WAE-based topic model; (iv) **ETM** (Dieng et al., 2020), a neural topic model which models the topic-word distribution matrix with word and topic embeddings; (v) **NSTM** (Zhao et al., 2021b), using the optimal transport distance to measure reconstruction error. (vi) **WeTe** (Wang et al., 2022), following NSTM and using conditional transport distance as reconstruction error.

## 4.2 MAIN RESULT OF TOPIC AND DOC-TOPIC DISTRIBUTION QUALITY

Table 2 reports the topic quality results concerning $C_V$ and TD, and Table 3 summarizes doc-topic distribution quality results concerning Purity and NMI of document clustering. We have the following observations: (i) **ECRTM effectively addresses the topic collapsing issue and outperforms baselines in topic quality.** In Table 2, the much lower TD scores imply baselines generate repetitive topics and thus suffer from topic collapsing. As aforementioned, these repetitive topics are less useful for downstream tasks and damage the interpretability of topic models. In contrast, we

Table 4: Ablation study. While DKM, DKM + Entropy, w/o ECR all have low TD, our ECRTM achieves much higher TD with better Purity and NMI. The low TD scores mean most topics are repetitive and less useful, making high $C_V$ scores meaningless. ‡ means the gain of ECRTM is statistically significant at 0.05 level.

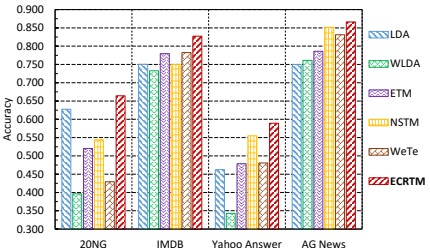

Figure 4: Text classification results. The improvements of ECRTM are all statistically significant at 0.01 level.

| Model | 20NG | | | | Yahoo Answer | | | |
|---|---|---|---|---|---|---|---|---|
| | Purity | NMI | $C_V$ | TD | Purity | NMI | $C_V$ | TD |
| DKM | ‡0.510 | ‡0.471 | 0.448 | ‡0.577 | ‡0.507 | ‡0.282 | 0.403 | ‡0.631 |
| DKM + Entropy | ‡0.222 | ‡0.148 | 0.469 | ‡0.503 | ‡0.252 | ‡0.092 | 0.433 | ‡0.592 |
| w/o ECR | ‡0.504 | ‡0.446 | 0.461 | ‡0.548 | ‡0.498 | ‡0.262 | 0.435 | ‡0.608 |
| **ECRTM** | 0.560 | 0.524 | 0.431 | 0.964 | 0.550 | 0.295 | 0.405 | 0.985 |

see our ECRTM achieves significantly higher TD scores across all datasets and mostly the best $C_V$ scores meanwhile. We emphasize although the $C_V$ of ECRTM is slightly higher than NSTM (0.389 v.s. 0.387) on Yahoo Answer, ECRTM completely outperforms on TD (0.903 v.s. 0.659). These results demonstrate that ECRTM produces more coherent and diverse topics than state-of-the-art baselines. These improvements are because our ECRTM makes topic embeddings away from each other and cover different semantics of word embeddings in the space instead of collapsing together as some baselines. (**ii**) **ECRTM surpasses baselines in inferring high-quality doc-topic distributions.** Table 3 shows our ECRTM consistently outperforms the baseline models by a large margin in terms of Purity and NMI. For example, ECRTM reaches 0.560 and 0.524 for Purity and NMI on 20NG, while the runner-up only has 0.354 and 0.356. These manifest that ECRTM not only achieves higher-quality topics but also better doc-topic distributions as document representations. See Appendices A.3 and A.4 for topic examples and more visualizations.

### 4.3 ABLATION STUDY

We conduct ablation studies and show the necessity of our proposed Embedding Clustering Regularization (ECR). Specifically, we remove the ECR from our ECRTM, denoted as w/o ECR. We also compare with the state-of-the-art deep clustering method, DKM (Fard et al., 2020) and DKM with minimizing entropy (DKM + Entropy, see Sec. 3.3). Table 4 shows DKM, DKM + Entropy, and w/o ECR all suffer from topic collapsing as indicated by their much lower TD scores. Although they have high $C_V$, their terrible TD scores mean most topics are repetitive and less useful for downstream tasks, making the high $C_V$ scores meaningless (see examples in Appendix A.3 for illustrations). Conversely, our ECRTM improves TD scores by a large margin and achieves the best document clustering performance with much higher Purity and NMI. This is because our ECR, as an effective regularization, can avoid the collapsing of topic embeddings while DKM, DKM + Entropy, and w/o ECR cannot. These results demonstrate our ECR is effective and necessary to address the topic collapsing issue and achieve robust topic modeling performance.

### 4.4 TEXT CLASSIFICATION

To evaluate extrinsically, we further conduct text classification experiments as downstream tasks. Specifically, we use the doc-topic distributions learned by topic models as document features and train SVMs to predict the class of each document. As reported in Figure 4, ECRTM significantly outperforms baseline models on all datasets. These results demonstrate that our ECRTM can be better utilized in the classification downstream task.

## 5 CONCLUSION

In this paper, we propose the novel Embedding Clustering Regularization Topic Model (ECRTM) to address the topic collapsing issue. ECRTM learns topics under the new Embedding Clustering Regularization that forces each topic embedding to be the center of a separately aggregated word embedding cluster. Extensive experiments demonstrate that ECRTM successfully alleviates topic collapsing and consistently achieves state-of-the-art performance in terms of producing high-quality topics and topic distributions of documents. We hope our work can contribute to building more reliable topic modeling applications.

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

# A APPENDIX

Table 5: Top 10 related words of discovered topics from IMDB. Repetitive words are underlined.

| | |
|---|---|
| ETM | like better good especially end look much done way just
like just one way made much times really even feel
one like around sort looking kind good main look just |
| NSTM | just show even come time one good really going know
just even really something come going like actually things get
just one even something come way really like always good |
| WeTe | hitchcocks couldve wouldve shouldve familys wifes anyones everyones itll hollywood
hitchcocks anyones wouldve everyones wifes couldve familys itll shouldve hollywood
hitchcocks couldve wouldve shouldve wifes familys anyones everyones itll hollywood |
| DKM | christmas disney musical songs bill timeless prince art rock holiday
christmas santa childrens holiday betty age ann adult children toy
fantasy christmas magic effects magical santa special holiday childrens child |
| DKM + Entropy | funny day physical semi ever way old due seen zone
funny ever day old seen way physical due semi relationship
funny ever day semi seen physical way old due psychological |
| **ECRTM** | jackie martial chan kung arts kong hong stunts bruce fight
nominated nancy academy award awards oscar oscars jake nomination dracula
vampires vampire freddy zombies zombie nightmare serial halloween killer slasher |

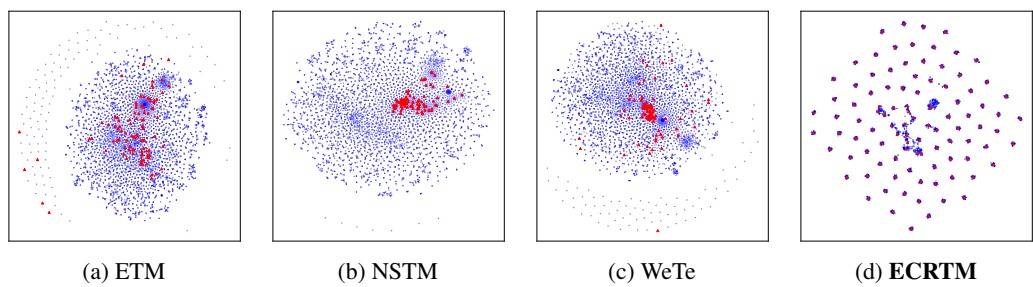

(a) ETM          (b) NSTM          (c) WeTe          (d) **ECRTM**

Figure 5: t-SNE (van der Maaten & Hinton, 2008) visualization of word embeddings (•) and topic embeddings (▲) under 100 topics. Topic embeddings commonly collapse together in state-of-the-art models (ETM (Dieng et al., 2020), NSTM (Zhao et al., 2021b), and WeTe (Wang et al., 2022)). In contrast, ECRTM can avoid the collapsing by forcing each topic embedding to be the center of a separately aggregated word embedding cluster.

## A.1 DATASET

We follow the dataset preprocessing steps of Card et al. (2018): (1) documents are tokenized and converted to lower case; (2) remove punctuation; (3) remove tokens that include numbers; (4) remove tokens less than 3 characters; (5) remove stop words.

## A.2 IMPLEMENTATION DETAIL

For pre-trained word embeddings, we employ 200-dimensional GloVe (Pennington et al., 2014). For the Sinkhorn's algorithm of ECRTM, we set the maximum number of iterations as 1,000, the stop tolerance 0.005 and $\varepsilon$ 0.05 following Cuturi (2013). For ECRTM, the prior distribution is specified

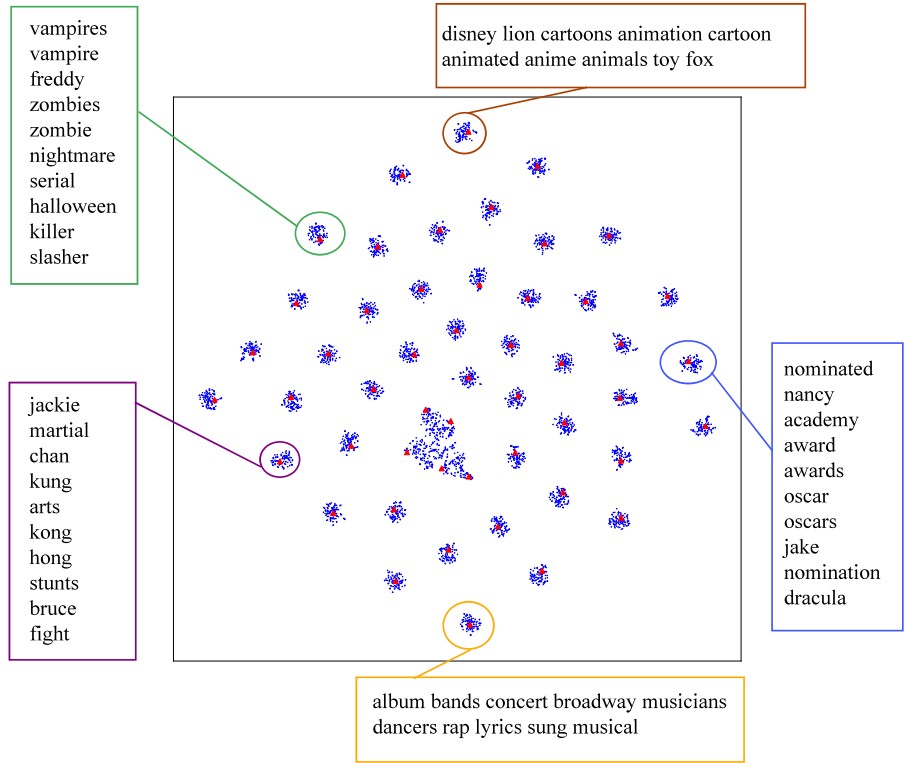

Figure 6: Visualization of discovered topics in the semantic space.

with Laplace approximation (Hennig et al., 2012) to approximate a symmetric Dirichlet prior as $\mu_{0,k} = 0$ and $\Sigma_{0,kk} = (K-1)/(\alpha K)$ with hyperparameter $\alpha$. The $\alpha$ in the prior distribution is $1.0$ following Card et al. (2018). Besides, $\tau$ is $0.2$, and $\lambda_{\text{ECR}}$ is $250$, $100$, $60$ and $5$ for 20NG, IMDB, Yahoo Answer and AG News respectively. Our encoder network is the same as Srivastava & Sutton (2017): a MLP that has two linear layers with softplus activation function, concatenated with two single layers each for the mean and covariance matrix. We use Adam optimizer (Kingma & Ba, 2014) to optimize the model parameters.

A.3 EXAMPLES OF DISCOVERED TOPICS

Table 5 shows randomly selected examples of discovered topics by different models from IMDB. We observe that ETM and NSTM both have highly uninformative and similar topics including common words like "just", "like", or "something". WeTe produces some exactly the same topics with words "couldve", "wouldve", and "shouldve". DKM and DKM + Entropy also generate repetitive topics with words "chirstmas", "holiday", and "funny". We see the topic collapsing issue commonly exists in these methods. These collapsed topics are uninformative and redundant, which are less useful for downstream applications and damage the interpretability of topic models. In contrast, the topics discovered by ECRTM are more distinct instead of repeating each other. Besides, they are relatively more coherent, such as the first topic with relevant words like "jackie", "chan", and "stunts". These examples show that our ECRTM generates higher-quality topics.

A.4 VISUALIZATION OF EMBEDDINGS

We visualize the learned topic and word embeddings with t-SNE (van der Maaten & Hinton, 2008) under 100 topics (Figure 1 is under 50 topics). Figure 5 shows while the topic embeddings mostly collapse together in the state-of-the-art baselines, our ECRTM avoids the collapsing of topic embeddings by forcing each topic embedding to be the center of a separately aggregated word embedding cluster. This illustrates that our ECR works effectively under a larger number of topics.

We further annotate the semantic space with the top related words of discovered topics by ECRTM as shown in Figure 6. We see that each word embedding cluster represents a diverse and coherent topic. This verifies that our ECRTM effectively clusters the embeddings of coherent words by jointly optimizing the neural topic modeling objective with the embedding clustering regularization objective.

## A.5 RUNTIME ANALYSIS

We train our model with NVIDIA GPU and use PyTorch for implementations. It takes about less than 0.5 GPU hours to train our model on datasets.

