# OpenReview forum: "Neural Topic Modeling with Embedding Clustering Regularization"
_ICLR.cc/2023/Conference — Submitted to ICLR 2023_

### Official Review · Reviewer_iecT · 2022-10-17

**Confidence:** 4
**Correctness:** 3
**Technical Novelty And Significance:** 2
**Empirical Novelty And Significance:** 2
**Recommendation:** 3

**Clarity, Quality, Novelty And Reproducibility:**

The paper is reasonably well written, even though there is some redundancy while some parts should have been a little bit more explained to make the paper more self-contained (e.g., the quick reference to Canas & Rosasco to explain Eq.2).

The paper is very focused on NTM approaches with a lapidary related work section. Many previous works have tried to leverage word embeddings and some of them may integrate the section. For instance, the work of Das et al. (https://aclanthology.org/P15-1077.pdf), who model the word embedding as a realization of a gaussian centered at the topic vector, seems really related to me.

The new regularization term, based on a clustering approach, looks quite new, even though it's used to address a problem that has been introduced by the neural models. However, the setup seems to need the addition of several "tricks" to make the whole thing works.

I didn't find any code for allowing an easy reproduction of the experiments. The pseudo code is given, so it should be possible to do it by yourself... maybe.


**Details Of Ethics Concerns:**

-

**Strength And Weaknesses:**

Strength:
- Adding a regularization term, but also other mechanisms (such as a priori on topic size), improves the results on some datasets

Weaknesses:
- Several mechanisms need to be combined to design the final model (clustering with soft assignment, prior on topic size, reintroduction of the "less sparse beta"). To this extent, stacking various techniques to get the final system introduces some doubt in what really explains the success of the approach. The ablation study of 4.3 doesn't really answer to this concern.
- As in previous approaches (which usually seem to overlook more than 20 years of research in topic model), the experimental framework is questionable. First, the perplexity cannot be discarded as easily: it is clearly not sufficient to assess the quality of topics, but it can be added to check whether the topics can explain/cover the full dataset. It's related to argument (2) p.2 and the way the authors address it is not convincing to me, even though it has already been used in previous ICRL papers. We shouldn't confuse topic modeling and clustering, many papers have been written on that subject. Evaluating the capacity of a topic model to cluster documents into a limited number of quite well-separated clusters (which is the case in most 20 newsgroups classes) is clearly not sufficient to evaluate the doc-topic distribution quality. Another remark: the C_V measure of Röder is not that much used in the literature and a couple of concerns (in term of reproducibility) have been raised online. I suggest to use UCI or UMass instead.
- Recent topic models should be able to deal with more diverse datasets than news and review.

**Summary Of The Paper:**

This work presents (yet another) Neural Topic Model (NTM). The main idea is to add a regularization term to force the topic vectors to be away in the space by using a clustering mechanism, which lead to decrease the collapsing problem. The full model, named Embedding Clustering Regularization Topic Model (ECRTM) is based on previous works following an Optimal Transport scheme. It is favorably compared to previous models on a set of classic (and limited) datasets.

**Summary Of The Review:**

This paper doesn't bring any big improvement in the topic modeling area. The regularization term may be somewhat useful in order to avoid topic collapsing, but the experiments fall short to really prove that the new model will be effective in a large range of dataset profiles.

---

> ### Author Response · Authors · 2022-11-16
> **Response**
>
>
> Thank you for your detailed reviews!
> We're happy you believe our method works. We note that our code has been submitted for reproducibility.
> We sincerely hope our responses can address your concerns and improve your evaluation. We're happy to answer your further questions.
>
> ===================================
> Q1: concerns about the ablation study since several mechanisms need to be combined
> ===================================
> Thank you for your question!
> We clarify the ablation study (Sec4.3) is sufficient to show the necessity of our method. Comparisons with w/o ECR show our clustering regularization is necessary; comparisons with DKM and DKM+Entropy show the necessity of sparse soft-assignments and cluster size constraints of our ECR. These results together demonstrate all the requirements (Sec3.3) are important to address the topic collapsing issue. Besides, our ECR alone can fulfill all these requirements without any other tricks.
> We're glad to address your further concerns.
>
> ===================================
> Q2: perplexity results
> ===================================
> Thank you for your suggestion!
> We report perplexity results in the following table (here we don't include some neural topic models (WLDA, NSTM, WeTe) as their structures are inapplicable to the perplexity approximation with ELBO. Please see (Miao et al 2016, Srivastava and Sutton 2017, Nan et al 2019, and Zhao et al IJCAI2021a)).
> We see our ECRTM also achieves better perplexity results (lower is better). We'll add this to our revision.
>
> | |20NG|IMDB|Yahoo Answer|AG News|
> |:---:|:---:|:---:|:---:|:---:|
> |LDA|2044.6|2482.8|4637.1|9951.1|
> |ETM|2113.4|1911.8|4653.4|1518.7|
> |ECRTM|1896.9|1830.8|3244.0|1436.1|
>
> ===================================
> Q3: why evaluate doc-topic distributions with document clustering
> ===================================
> We note that document clustering has been conducted in many previous studies to evaluate doc-topic distributions ([1-5], Yan ICDM2014, Boyd-Graber 2017, Zhao ICLR2021b, Wang ICLR2022).
> Besides, we also include text classification experiments (Sec4.4) with doc-topic distributions as features. This can further evaluate the quality of doc-topic distributions.
>
> [1]SIGIR2013. Improving LDA topic models for microblogs via tweet pooling and automatic labeling.
> [2]WWW2013. A biterm topic model for short texts.
> [3]KDD2014. A dirichlet multinomial mixture model-based approach for short text clustering.
> [4]AAAI2015. Short and sparse text topic modeling via self-aggregation.
> [5]SIGIR2016. Topic modeling for short texts with auxiliary word embeddings.
>
> ===================================
> Q4: reproducibility of CV
> ===================================
> We note that CV can be easily reproduced by the tool (https://github.com/dice-group/Palmetto).
> Besides, (Roder et al WSDM2015) have shown CV is more consistent with human judgment than NPMI, UCI, and UMass with extensive experiments. In our experiments, we also find NPMI (UCI, UMass) is less consistent with human judgment: NPMI (UCI, UMass) tends to give much higher scores to trivial and less informative topics than coherent topics. For example, we see Topic#4, #5, #6 are much more coherent and informative than Topic#1, #2, #3 in the following table. But unfortunately NPMI (UCI, UMass) gives high scores to Topic#1, #2, #3 and much lower scores to Topic#4, #5, #6. In contrast, CV is more consistent with human judgment as it gives higher scores to Topic#4, #5, #6 (Note that we shouldn't directly compare the NPMI (UCI, UMass) score with the CV score of a topic because they are in different scales).
> Please see (Roder et al WSDM2015) for more extensive experiments. We'll add more explanations to the revision.
>
> |NPMI|UCI|UMass|CV|Top words of topics|
> |:----|:----|:----|:----|:----|
> |0.069|0.746|-0.900|0.272|Topic#1: way come actually make yet|
> |0.065|0.615|-0.967|0.272|Topic#2: fact even indeed though kind|
> |0.065|0.628|-3.367|0.255|Topic#3: really pretty something seem seems|
> | | | | | |
> |-0.076|-2.660|-8.196|0.497|Topic#4: vampires vampire freddy zombies zombie|
> |-0.075|-2.678|-9.032|0.476|Topic#5: disney lion cartoons animation cartoon|
> |-0.114|-3.367|-6.120|0.452|Topic#6: jackie martial chan kung arts|
>
>
> ===================================
> Q5: dataset diversity
> ===================================
> We clarify that our used datasets are quite diverse, including news, reviews, and Q&A. They're widely-used benchmark datasets for experiments. Please kindly reply to us if you believe any dataset is necessary.
>
> ===================================
> Q6: some missing references
> ===================================
> Thank you for your suggestion! We'll add the important reference (Das et al 2015) and more others to the related work section.
>
> ===================================
> Q7: code release
> ===================================
> Thank you for mentioning this. We've submitted an anonymous link to our code in the comment.

---

### Official Review · Reviewer_1otX · 2022-10-24

**Confidence:** 4
**Correctness:** 2
**Technical Novelty And Significance:** 3
**Empirical Novelty And Significance:** 3
**Recommendation:** 5

**Clarity, Quality, Novelty And Reproducibility:**

# Clarity

The paper is written very clearly and the best part is that the author(s) address the limitations or
future research opportunities in this topic wherever necessary. Almost all topics discussed are easy to
follow and understand. This paper sufficiently covers the context of the topic through the mentioned literature which also
seems to be quiet relevant to the work. It also discusses how traditional topic modeling is different
than the recent neural topic models and how the clustering of word embeddings is different than
modeling topic-word distribution. Although these ideas and their differences are covered, not
all of them are presented very well. Some sections can be co-ordinated better and for any section the main ideas should be
discussed to the point

# Quality

- it is mentioned that it is difficult to use the Dirichlet distribution with neural topic models. This is a completely unfounded claim. There are more than five different ways of using Dirichlet priors in VAEs, one is implemented in Pytorch as default where you just have to use the Dirichlet, and the reparameterization is done automatically, and most of them are very easy to use. It has been shown in previous work that the Dirichlet prior gives much better results than the Gaussian one or the Laplace approximation.
- Overall quality of the paper with respect to the posterior collapse problem is fair. Some sections can be written more to the point by focusing on the central
idea of the section at the start of the para. For example, in section ”Fulfilling Preset Cluster Size
constraints” where formation of empty clusters seems to be the main agenda of the paragraph but it
is only discussed towards the end.

# Originality

The novelty in this paper is to produce sparse soft-assignments using Deep KMeans method (Deep
k-means: Jointly clustering with k-means and learning representations) by restricting each word em-
bedding to belong to only single topic embedding by pushing it closer to only ”one” topic embedding
and away from other topic embeddings while also constraining cluster sizes to be equal for all topics. The idea seems to be simple yet significant when tackling posterior collapse problem in topic models.
Although, it is unclear if such separation of topic-word clusters would lead to some other problems
such as latent holes which might produce undesirable results.

# Reproducibility

Code does not seem to be available. Otherwise, datasets and benchmarks are standard and the method is described well.

**Strength And Weaknesses:**

# Strengths

- Clear illustration of limitations of existing methods. It is also clearly illustrated that the proposed model
overcomes those limitations.
- A sound theoretical solution to the limitations is presented.
- Overall good results when compared to the selected baselines

#Weaknesses
- The baseline selection is poor. The authors claim that it is difficult to incorporate Dirichlet distribution
in VAEs. This is not entirely true. There has been work by (Burkhardt & Kramer, 2019) on using
Dirichlet as prior for VAEs in topic modeling. This work was again reimplemented by (Hoyle et al.,
2021). This has been overlooked by the authors in the paper. The compared baselines for illustrations of
limitations also do not use Dirichlet prior, which was initially presented to overcome the said limitations.
- The evaluation includes Cv score, which for some experiments is very low. It is recommended to include
NPMI scores as well.
- If I understand the sparse soft-assignment clustering correctly, this technique will assign every word
to one and only one topic cluster but there are several words that belong to multiple contexts for
example, season, which can be related to weather and sports both. So, the proposed model cannot
assign words that occur in different contexts to different topics.
- The most obvious limitation is that this method produces fixed size clusters of all topics. On page
5, section 3.4, they assume that every topic includes same amount of semantic information and this
assumption is supported by practically forcing every topic to have same number of words. This means
having same amount of semantic information is equal to having same number of words. I don’t quite
agree to this assumption because semantic information of a topic cannot be quantified by the number
of words present in it.
- Lastly, posterior collapse is a significant and long standing problem in several other text (and
image) applications such as text generation, summarization etc. The method performs considerably
well for topic modeling but unfortunately does not extend or discuss how it can be generalized to the
other applications.

**Summary Of The Paper:**

This paper proposes an embedding clustering regularization technique to mitigate the posterior collapse
problem in neural topic models. The regularization technique forces the topic embeddings to be the
centers of separately aggregated clusters of word embeddings. As a result, distinct clusters of topic-
word embeddings are formed wherein each topic comprises distinct words and every topic is distant
from one another in the semantic space. This paper evaluates their technique for different aspects
of topic modeling such as topic quality (using coherence) and diversity and in addition also measures
topic-doc distribution and document classification accuracy. It outperforms several other methods that
it compares with on almost every evaluation metric that they use.

**Summary Of The Review:**

The work in itself is novel and overcomes topic collapsing limitations. But, the baseline selection is incomplete.
Methodology is clearly explained. There are nice illustrations of limitations and results which suggest that
the proposed model is effective. However, without comparison to SotA baselines, it is difficult to evaluate the results.

Solving a notorious problem of posterior collapse in topic models is the main contribution of this paper.
Although, it is based on the prior clustering techniques such as Deep Kmeans, some small tweaks in
their methodology seems to significantly improve the topic collapsing issue and this paper successfully
demonstrates it via empirical setup. The qualitative results shown also seem to be promising. The
approach outperforms many recent topic models on several evaluation criteria such as topic quality
(coherence), diversity and downstream document classification task but there is still some scope for
rigorous experiments to demonstrate the robustness of the idea against different lengths of text, different
topic sizes and other downstream tasks such as document retrieval and also if it can be generalized for
other NLP tasks.




Open questions:
- What about the robustness of the model against the number of topics? (The results are shown only
for K=50,100)
- Co-occurence information between words is comparatively low for short text such as tweets or news
headlines. How does the method perform on short (short text dataset M10) text sequences?
- Any specific reason for not comparing the results with those from Zhang et al paper (Is Neural Topic
Modelling Better than Clustering? An Empirical Study on Clustering with Contextual Embeddings
for Topics)?

---

> ### Author Response · Authors · 2022-11-16
> **Response**
>
>
> Thank you for your helpful feedback!
> We are encouraged that you believe our paper is clear, our method is theoretically sound, and our results are good.
> We note that our code has been submitted for reproducibility.
> We sincerely hope our responses can address your concerns and improve your evaluation.
>
> ===================================
> Q1: robustness to the number of topics.
> ===================================
> Here we report the results on 20NG under different numbers of topics(K). We see our method is robust to the number of topics.
> Thank you for your question! We'll add these results to our revision.
>
> | |K=10| |K=20| |K=30| |K=40| |
> |:---:|:---:|:---:|:---:|:---:|:---:|:---:|:---:|:---:|
> | |CV|TD|CV|TD|CV|TD|CV|TD|
> |LDA|0.376|0.627|0.384|0.620|0.377|0.678|0.390|0.697|
> |KM|0.208|0.207|0.230|0.180|0.247|0.151|0.255|0.217|
> |WLDA|0.354|0.533|0.354|0.443|0.360|0.389|0.357|0.430|
> |ETM|0.380|0.820|0.372|0.763|0.373|0.744|0.383|0.698|
> |NSTM|0.397|0.600|0.381|0.487|0.385|0.391|0.389|0.418|
> |WeTe|0.422|1.000|0.380|0.980|0.387|0.980|0.388|0.978|
> |ECRTM|0.487|1.000|0.454|1.000|0.437|1.000|0.435|0.993|
>
> | |K=60| |K=70| |K=80| |K=90| |
> |:---:|:---:|:---:|:---:|:---:|:---:|:---:|:---:|:---:|
> | |CV|TD|CV|TD|CV|TD|CV|TD|
> |LDA|0.373|0.674|0.385|0.590|0.381|0.645|0.380|0.656|
> |KM|0.269|0.219|0.282|0.256|0.300|0.321|0.289|0.300|
> |WLDA|0.361|0.343|0.373|0.335|0.371|0.345|0.375|0.287|
> |ETM|0.367|0.672|0.377|0.667|0.377|0.663|0.372|0.629|
> |NSTM|0.392|0.444|0.396|0.520|0.400|0.468|0.386|0.450|
> |WeTe|0.378|0.948|0.368|0.879|0.355|0.800|0.349|0.754|
> |ECRTM|0.413|0.993|0.405|0.910|0.410|0.957|0.402|0.906|
>
> ===================================
> Q2: comparison with DVAE
> ===================================
> Thank you for your suggestion!
> The reason why we didn't include DVAE is that DAVE has been compared with NSTM by (Zhao et al ICLR2021b), and they find NSTM generally outperforms DVAE. In our paper we show our method can exceed NSTM.
> But we agree that a more inclusive experiment would help; thus we report the comparisons with DVAE as follows. We see our method also surpasses DVAE. We will add these to our revision.
>
> ||20NG| | | |IMDB| | | |Yahoo Answer| | | |AG News| | | |
> |:---:|:---:|:---:|:---:|:---:|:---:|:---:|:---:|:---:|:---:|:---:|:---:|:---:|:---:|:---:|:---:|:---:|
> | |K=50| |K=100| |K=50| |K=100| |K=50| |K=100| |K=50| |K=100| |
> | |CV|TD|CV|TD|CV|TD|CV|TD|CV|TD|CV|TD|CV|TD|CV|TD|
> |DVAE|0.331|0.598|0.372|0.658|0.294|0.050|0.290|0.229|0.338|0.674|0.376|0.589|0.419|0.347|0.298|0.113|
> |ECRTM|0.431|0.964|0.405|0.904|0.395|0.979|0.373|0.887|0.405|0.985|0.389|0.903|0.466|0.961|0.416|0.981|
>
> ===================================
> Q3: performance on short texts
> ===================================
> We note that we've already included short text datasets: Yahoo Answer and AG News. Their average document length is about 35 and 20 respectively. The experiments also show our method achieves state-of-the-art performance on short texts with sparse word co-occurrence information.
> Here we additionally report the results on the short text dataset M10 where our method also outperforms baselines:
>
> | |K=50| |K=100| |
> |:----|:----|:----|:----|:----|
> | |CV|TD|CV|TD|
> |LDA|0.352|0.958|0.372|0.851|
> |KM|0.218|0.135|0.250|0.196|
> |WLDA|0.372|0.427|0.373|0.389|
> |DVAE|0.419|0.258|0.419|0.541|
> |ETM|0.357|0.863|0.371|0.794|
> |NSTM|0.413|0.929|0.416|0.882|
> |WeTe|0.338|0.812|0.304|0.612|
> |ECRTM|0.426|1.000|0.426|0.988|
>
>
> ===================================
> Q4: Can one word belong to different topics
> ===================================
> Thank you for your insightful question!
> We note that in our model one word can belong to different topics.
> We only use sparse soft-assignments in the regularization, not for modeling topics. We model learned topics by formulating beta in Eq(4) instead of directly using the sparse soft-assignments. As discussed in Sec3.5, beta is much less sparse than soft-assignments. Thus it enables one word to belong to different topics with different weights. We've added more clarifications about this to avoid misunderstandings.
>
>
> ===================================
> Q5: solve posterior collapse issue in VAE of other applications
> ===================================
> Thank you for your question!
> We agree that it would be great if some intuitions or techniques can be generalized to the posterior collapse of VAE. But unfortunately it may be a much more difficult topic and beyond the scope of this work.

---

> > ### Author Response · Authors · 2022-11-16
> > **Response**
> >
> >
> > ===================================
> > Q6: CV is used in experiments. include NPMI as well
> > ===================================
> > Thank you for your suggestion!
> > We note that (Roder et al WSDM2015) have shown CV is more consistent with human judgment than NPMI, UCI, and UMass with extensive experiments.
> > In our experiments, we also find NPMI (UCI, UMass) is less consistent with human judgment: NPMI (UCI, UMass) tends to give much higher scores to trivial and less informative topics than coherent topics. For example, we see Topic#4, #5, #6 are much more coherent and informative than Topic#1, #2, #3 in the following table. But unfortunately NPMI (UCI, UMass) gives high scores to Topic#1, #2, #3 and much lower scores to Topic#4, #5, and #6. In contrast, CV is more consistent with human judgment as it gives higher scores to Topic#4, #5, #6 (Note that we shouldn't directly compare the NPMI (UCI, UMass) score with the CV score of a topic because they are in different scales).
> > Please see (Roder et al WSDM2015) for more extensive experiments. We'll add more explanations to the revision.
> >
> > |NPMI|UCI|UMass|CV|Top words of topics|
> > |:----|:----|:----|:----|:----|
> > |0.069|0.746|-0.900|0.272|Topic#1: way come actually make yet|
> > |0.065|0.615|-0.967|0.272|Topic#2: fact even indeed though kind|
> > |0.065|0.628|-3.367|0.255|Topic#3: really pretty something seem seems|
> > | | | | | |
> > |-0.076|-2.660|-8.196|0.497|Topic#4: vampires vampire freddy zombies zombie|
> > |-0.075|-2.678|-9.032|0.476|Topic#5: disney lion cartoons animation cartoon|
> > |-0.114|-3.367|-6.120|0.452|Topic#6: jackie martial chan kung arts|
> >
> >
> >
> >
> > ===================================
> > Q7: Why all cluster sizes are constrained to be the same
> > ===================================
> > We note that we propose cluster size constraints to avoid the trivial solutions of empty clusters. Our method is compatible with various cluster size constraints. But unfortunately we usually don't have reliable prior information to customize the size of each cluster for a specific dataset. Therefore we simply constrain the cluster sizes to be the same and find it already works pretty well.
> > We agree that how to specify the cluster sizes for a dataset may be an interesting future direction and will leave it for future exploration.
> > Thank you for your comment! We've removed the misunderstandings about this part.
> >
> >
> > ===================================
> > Q8: why not compare with (Zhang et al 2022)
> > ===================================
> > Thank you for your question!
> > (Zhang et al 2022)'s work is quite inspiring. We note it's unfair to compare with (Zhang et al 2022) since they additionally use high-quality contextualized sentence embeddings while NTM baselines can only use word embeddings for modeling topics.
> > Besides, we've fairly compared with KM (Sia et al 2020, similar idea to (Zhang et al 2022)) with the same word embeddings as other baselines. The experiments show our method outperforms KM by a large margin.

---

### Official Review · Reviewer_YKAw · 2022-10-27

**Confidence:** 4
**Correctness:** 3
**Technical Novelty And Significance:** 2
**Empirical Novelty And Significance:** 3
**Recommendation:** 6

**Clarity, Quality, Novelty And Reproducibility:**

Quality:
- Topics in Table-1 do not look very informative and coherent. So, is the issue of topic collapsing present in coherent and incoherent topics alike or is it skewed?
- The proposed ECR objective generates sparse soft-assignments of word embeddings to topic embeddings which makes sure that each word embedding only belongs to one topic mainly. So, how this method will deal with the words having multiple semantic meanings like "bank" (river bank, financial bank), "chip" (potato chip, electronic chip), "apple" (fruit, company) etc. belonging to multiple topics?
- How is the performance of ECRTM using contextualized word embeddings like BERT, for initialization of word embeddings matrix, to correctly identify the semantics of ambiguous words compare with non-contextualized word embeddings like GloVe?
- Table-5 (Qualitative analysis): Comparison of 3 unrelated topics of baseline methods and proposed method is uninformative. It would be better to compare all 50 topics (top 5 or 10 words for K = 50) of any baseline method and the proposed method for a better picture regarding topic collapsing and coherency. With such full comparison of all topics it can be checked if good quality and coherent topics which are present in the topics extracted using baseline models are absent or present in the topics extracted using proposed models.

Clarity:
- Typographical errors:
	- [Section 3.2, 2nd paragraph] word distribution x -> word distribution beta
- A figure of the proposed ECRTM model with joint optimization objective would boost understanding of the model architecture and working.
- A table of notations would improve readability of the mathematical formulation.

Novelty:
- Related research has already worked on clustering of word embeddings to extract topics. Although they are not technically topic models, the inspiration for word embedding clustering to extract coherent and diverse topics can be derived. However, the issue of topic collapsing is very critical and the proposed method is promising and should be utilized in future works on neural topic modeling.
- How is this work compared to Orthogonal Topic modeling [1] ?

Reproducibility:
- Experimental setup and hyperparameter settings are described in detail.


**Strength And Weaknesses:**

Strengths:
- The motivation is well founded and the claims are sound.
- Paper is clearly presented and easy to follow.
- Mathematical formulation is explanatory and easy to understand.
- Proposed ECR objective is robust and effective.
- Experimental evaluation is extensive and the proposed topic model consistently outperforms multiple NTM baselines on topic quality and topic diversity metrics with significant margins over multiple datasets.

Weaknesses:
- there are limitations to the proposed approach, including multi-sense/ambiguous words in topics and comparisons to traditional baselines such as Orthogonal topic modeling.
- Contributions are incremental and Novelty is limited.
- missing baselines and experimental comparisons such as Orthogonal topic modeling and neural topic models such as DocNADE or iDocNADEe [2]
- topic analysis (qualitative) is unclear and seems biased to initial (3) set of topics - however should be analyzed for all topics extracted (extend the appendix)

Question:
- how is this work compared to Orthogonal Topic Modeling?

References:
1. Probabilistic Text Modeling with Orthogonalized Topics. SIGIR 2014.
2. Document Informed Neural Autoregressive Topic Models with Distributional Prior. AAAI 2019.


**Summary Of The Paper:**

- This paper tackles the well known and widely observed issue of topic collapsing in neural topic models (NTMs).
- Specifically, it proposes a novel embedding clustering regularization (ECR) objective, in addition to reconstruction error objective, during training variational autoencoder (VAE) based NTM which separately aggregates the word embeddings in clusters and apply constraint on topic embeddings to be within those clusters.
- A new topic model is further proposed which jointly optimizes using reconstruction error objective and the ECR objective to extract non-collapsing, diverse and coherent topics.
- the paper takes inventive steps in addressing topic collapsing in neural topic modeling - however there are limitations, including multi-sense/ambiguous words in topics and comparisons to traditional baselines such as Orthogonal topic modeling.



**Summary Of The Review:**

- Although novelty is limited in this paper, but the direction it explores to eliminate topic collapsing issue and generate diverse and coherent topics is interesting and important as diverse topics generate high quality document-topic representations which further boost accuracy in downstream supervised or unsupervised NLP tasks.
- missing comparison to (related) baselines methods (please see weaknesses)
- additionally, please see weaknesses section

---

> ### Author Response · Authors · 2022-11-16
> **Response**
>
>
> Thank you for your insightful comments!
> We're encouraged that you believe our paper is clear, our method is theoretically sound, and our results are good.
> We sincerely hope our responses can address your concerns and improve your evaluation.
>
>
>
> ===================================
> Q1: Can one word belong to different topics
> ===================================
> Thank you for your insightful question!
> We note that in our model one word can belong to different topics.
> We only use sparse soft-assignments in the regularization, not for modeling topics. We model learned topics by formulating beta in Eq(4) instead of directly using the sparse soft-assignments. As discussed in Sec3.5, beta is much less sparse than soft-assignments. Thus it enables one word to belong to different topics with different weights. We've added more clarifications about this to avoid misunderstandings.
>
>
> ===================================
> Q2: compare with orthogonal topic modeling and DocNADE
> ===================================
> Thank you for your mentioning these important work! We'll add these references and we've started experiments to compare our method with them.
> Moreover, we'd like to emphasize we've compared with the most recent state-of-the-art work like ETM (Dieng et al TACL2020), NSTM (Zhao et al ICLR2021b), and WeTe (Wang et al ICLR2022). The results show our model significantly outperforms them and effectively solves the topic collapsing issue.
>
>
> ===================================
> Q3: try contextualized word embeddings
> ===================================
> Thank you for your constructive suggestion!
> Currently we need to follow previous work and use non-contextualized word embeddings for fair comparisons.
> But we agree that this is a promising direction, and we've started some experiments and will add them to the revision if we get some interesting results.
>
>
>
> ===================================
> Q4: list all discovered topics in Table5
> ===================================
> Thank you for your comment!
> We note that Table5 is only a case study with some examples following plentiful early work (Nan et al ACL2019, Dieng et al TACL2020, Zhao et al ICLR2021b, Wang et al ICLR2022).
> But we agree with your comment and we will extend Table5 in the appendix.

---

### Official Review · Reviewer_QM6u · 2022-11-03

**Confidence:** 3
**Clarity, Quality, Novelty And Reproducibility:** 1. Why the results of DKM and DKM+ent…
**Correctness:** 3
**Technical Novelty And Significance:** 3
**Empirical Novelty And Significance:** 3
**Recommendation:** 6

**Strength And Weaknesses:**

Strength:
The proposed method regularizes the topic model such that the topic embeddings are located at the center of word embedding clusters. This is demonstrated in the t-sne Figure. Since the clusters are separated, we can observe improvements in terms of TD in both Table 2 and Table 3.

Weakness:
1. As the ECR is proposed as an additional module to further regularize the topic models, it will be great to also apply the ECR to other baseline approaches.
2. It will be nice to have some proof or analysis to explain the necessity of forcing the topic embeddings to be the center of the word embedding clusters.
3. More experiments can be conducted to enhance the current findings. Such as running the experiments with different K, ranging from 10 to 100 with a fixed step size.


**Summary Of The Paper:**

The author found that the previous methods often suffer topic collapsing issues, which leads to repetitive topics, insufficient topic discovery, and also damages the model interpretability. Motivated by such findings, the paper proposed to add Embedding Clustering Regularization (ECR) to topic modeling models. The ECR helps to keep the topic embedding at the center of word embedding clusters and then alleviates the topic collapsing issue. The proposed method is evaluated on four benchmark datasets and the experimental results show encouraging performance.

**Summary Of The Review:**

The paper proposed to add Embedding Clustering Regularization (ECR) to topic modeling models. The ECR helps to keep the topic embedding at the center of word embedding clusters and then alleviates the topic collapsing issue. The proposed method is evaluated on four benchmark datasets and the experimental results show encouraging performance.

---

> ### Author Response · Authors · 2022-11-16
> **Response**
>
>
> Thank you for your helpful reviews!
> We're happy that you appreciate our reasonable method and experimental results.
> We sincerely hope our responses can address your concerns and improve your evaluation.
>
>
> ===================================
> Q1: applying proposed ECR to other baselines
> ===================================
> Thank you for the suggestion!
> We actually tried applying ECR to some baselines.
> We add our ECR to the simplest baseline without any tricks, which can achieve state-of-the-art performance. This serves as an ablation study to show the effectiveness of our ECR. We find it may be unsuitable to add ECR directly to some other baselines like NSTM and WeTe due to their specially designed structures, so we plan to leave it for future exploration.
>
>
> ===================================
> Q2: the necessity of regularization on topic embeddings
> ===================================
> We note that we've explained the necessity of regularization on topic embeddings in Sec3.2 and Sec3.3, and shown with experiments in Sec4.3.
> In Sec3.2, we show the topic collapsing issue is due to the collapsing of topic embeddings for previous methods. To avoid this collapsing, it's necessary to make topic embeddings away from each other and cover different semantics of word embeddings. This motivates the regularization on topic embeddings in Sec3.3. Experiments in Sec4.3 show DKM, DKM+Entropy, and w/o ECR all fail to properly regularize topic embeddings while our ECRTM can. The superior performance of ECRTM can demonstrate the necessity of the regularization on topic embeddings.
>
>
> ===================================
> Q3: robustness to different numbers of topics
> ===================================
> Here we report the results on 20NG under different numbers of topics (K). We see our method is robust to the number of topics.
> Thank you for your question! We'll add these results to our revision.
>
>
> | |K=10| |K=20| |K=30| |K=40| |
> |:---:|:---:|:---:|:---:|:---:|:---:|:---:|:---:|:---:|
> | |CV|TD|CV|TD|CV|TD|CV|TD|
> |LDA|0.376|0.627|0.384|0.620|0.377|0.678|0.390|0.697|
> |KM|0.208|0.207|0.230|0.180|0.247|0.151|0.255|0.217|
> |WLDA|0.354|0.533|0.354|0.443|0.360|0.389|0.357|0.430|
> |ETM|0.380|0.820|0.372|0.763|0.373|0.744|0.383|0.698|
> |NSTM|0.397|0.600|0.381|0.487|0.385|0.391|0.389|0.418|
> |WeTe|0.422|1.000|0.380|0.980|0.387|0.980|0.388|0.978|
> |ECRTM|0.487|1.000|0.454|1.000|0.437|1.000|0.435|0.993|
>
> | |K=60| |K=70| |K=80| |K=90| |
> |:---:|:---:|:---:|:---:|:---:|:---:|:---:|:---:|:---:|
> | |CV|TD|CV|TD|CV|TD|CV|TD|
> |LDA|0.373|0.674|0.385|0.590|0.381|0.645|0.380|0.656|
> |KM|0.269|0.219|0.282|0.256|0.300|0.321|0.289|0.300|
> |WLDA|0.361|0.343|0.373|0.335|0.371|0.345|0.375|0.287|
> |ETM|0.367|0.672|0.377|0.667|0.377|0.663|0.372|0.629|
> |NSTM|0.392|0.444|0.396|0.520|0.400|0.468|0.386|0.450|
> |WeTe|0.378|0.948|0.368|0.879|0.355|0.800|0.349|0.754|
> |ECRTM|0.413|0.993|0.405|0.910|0.410|0.957|0.402|0.906|
>
>
>
> ===================================
> Q4: why DKM and DKM+Entropy are in the ablation study
> ===================================
> We note that DKM and DKM+Entropy are NOT previously proposed topic modeling baselines. We're the first to propose them as possible options for the regularization in our work. Thus we compare them in the ablation study section to show the necessity of our new ECR method.
>
>
> ===================================
> Q5: DKM has higher topic coherence
> ===================================
> We'd like to emphasize that the poor topic diversity of DKM makes its high coherence less meaningful.
> Specifically, DKM repeats coherent topics to trickly improve coherence but still suffers from topic collapsing: the topics are highly repetitive and less informative, as exemplified in Table5. In consequence, these topics hinder the understanding of documents and are less useful to downstream applications. This issue has been confirmed in several previous studies (Nan et al ACL2019, Wu et al EMNLP2020b, Dieng et al 2019, Dieng et al TACL2020).
> In contrast, our ECRTM achieves both high coherence and high diversity, and significantly better document clustering performance (Purity and NMI).
>
>
> ===================================
> Q6: Why choose a large K instead of a small one like the number of labels
> ===================================
> We note that topic modeling studies commonly use a relatively large K (number of topics) for sufficient evaluation (Miao et al ICML2016, Srivastava and Sutton ICLR2017, Nan et al ACL2019, Zhao et al ICLR2021b, Wang et al ICLR2022).
> This is because it could be difficult to compare the quality of topics with a very small K. For example if we only discover one topic, the topic diversity is always perfect in every model. So we cannot compare different models.
> With a large number of topics, we can compare the quality of many topics to sufficiently evaluate the performance.
> Besides, our method is robust to the number of topics (ranging from small to large) as reported in Q3.

---

### Decision · Program_Chairs · 2023-01-20

**Decision:**

Reject

**Justification For Why Not Higher Score:**

None of the referees raised their scores. Also given that Topic modelling is a well researched area, current results does not seem to make
too much progress

**Justification For Why Not Lower Score:**

N/A

**Metareview: Summary, Strengths And Weaknesses:**

The paper proposes a novel neural Topic model ECRTM which is claimed to resilient to the problem of Topic Collapse.
It argues that a clustering based regularisation will yield topics that are more well-separated and hence mitigate the problem of Topic collapse.
It sets up an objective which augments the loss by an additive term to enforce the regularisation.

The paper is clearly written and the main claims are well communicated. Experimental evaluation shows that the proposed method outperforms chosen baselines.

Given that Topic modelling is a widely researched area over last two years the proposed novelty is modest.
Concerns are raised about selection of baselines. Measures such as perplexity were not reported.

A more rigorous characterisation of separability of Topics may help the claims.



**Summary Of Ac-Reviewer Meeting:**

This is a clear reject paper and hence the meeting was not needed. The average rating was 5 which is below the suggested guideline, and also s the novelty seems to be incremental.